# ANALOGICAL REASONING FOR VISUALLY GROUNDED COMPOSITIONAL GENERALIZATION

## ABSTRACT

Children acquire language subconsciously by observing the surrounding world and listening to descriptions. They can discover the meaning of words even without explicit language knowledge, and generalize to novel compositions effortlessly. In this paper, we bring this ability to AI, by studying the task of multimodal compositional generalization within the context of visually grounded language acquisition. We propose a multimodal transformer model augmented with a novel mechanism for analogical reasoning, which approximates novel compositions by learning semantic mapping and reasoning operations from previously seen compositions. Our proposed method, Analogical Reasoning Transformer Networks (ARTNET), is trained on raw multimedia data (video frames and transcripts), and after observing a set of compositions such as "washing apple" or "cutting carrot", it can generalize and recognize new compositions in new video frames, such as "washing carrot" or "cutting apple". To this end, ARTNET refers to relevant instances in the training data and uses their visual features and captions to establish analogies with the query image. Then it chooses a suitable verb and noun to create a new composition that describes the new image best. Extensive experiments on an instructional video dataset demonstrate that the proposed method achieves significantly better generalization capability and recognition accuracy compared to state-of-the-art transformer models.

## 1 INTRODUCTION

Visually grounded Language Acquisition (VLA) is an innate ability of the human brain. It refers to the way children learn their native language from scratch, through exploration, observation, and listening (*i.e.,* self-supervision), and without taking language training lessons (*i.e.,* explicit supervision). 2-year-old children are able to quickly learn semantics of phrases and their constituent words after repeatedly hearing phrases like "washing apple", or "cutting carrot" and observing such situations. More interestingly, they will also understand new compositions such as "washing carrot" or "cutting apple", even before experiencing them. This ability of human cognition is called compositional generalization (Montague (1970); Minsky (1988); Lake et al. (2017)). It helps humans use a limited set of known components (vocabulary) to understand and produce unlimited new compositions (*e.g.* verb-noun, adjective-noun, or adverb-verb compositions). This is also one of the long-term goals of Artificial Intelligence (AI), *e.g.* in robotics, where it enables the robot to learn new instructions that they have never heard before.

Nevertheless, contemporary machine intelligence needs to overcome several major challenges of the task. On one hand, learning compositional generalization can be difficult without using data-hungry models. The power of existing language models mainly rely on large-scale language corpora (Lake & Baroni (2017); Pennington et al. (2014); Devlin et al. (2018)). They are still inadequate at compositional generalization (Marcus (1998); Lake & Baroni (2018); Surís et al. (2019)). Their goal is to recognize training examples rather than focusing on what is missing from training data. On the other hand, the designed model should close the paradigmatic gap (Nikolaus et al. (2019)) between seen compositions and new compositions. For instance, given seen verb-noun compositions "1A" and "2B" (the digit indicates verb, the letter indicates noun), the model should be able to link seen compositions to new compositions (like "1B" or "2A") in completely new cases.

Figure 1: We propose a multimodal language acquisition approach inspired by human language learning processes consisting of three steps: association, reasoning, and inference.

Different from previous work (Johnson et al. (2017); Baradel et al. (2018); Santoro et al. (2017)), we bring the power of compositional generalization to state-of-the-art language models by incorporating Analogical Reasoning (AR) (Gentner & Smith (2012); Littlemore (2008); Vosniadou & Ortony (1989)). An analogy is a comparison between similar concepts or situations, and AR is analogical semantic reasoning that relies upon an analogy. The human brain spontaneously engages in AR to make sense of unfamiliar situations in every day life (Vamvakoussi (2019)). Inspired by the AR process in the human brain, we design the counterpart for machine language acquisition. To this end, we create a language model that generate appropriate novel compositions by relevant seen compositions, and forming analogies and appropriate arithmetic operations to express the new compositions (*e.g.* "washing carrot" = "washing apple' + "cutting carrot" - "cutting apple"). We describe this process in three steps: association, reasoning, and inference, as shown in Figure 1.

Given an image (a video frame in our case) and a narrative sentence describing it, we mask the main verb-noun composition from the sentence, and ask the model to guess the correct composition that completes the sentence, considering the provided image. To this end, we propose a novel self-supervised and reasoning-augmented framework, Analogical Reasoning Transformer Networks (ARTNET). ARTNET adopts a multimodal transformer (similar to ViLBERT (Lu et al. (2019))) as its backbone to represent visual-textual data in a common space. Then it builds three novel modules on top of the backbone that corresponds to the aforementioned AR steps: association, reasoning, and inference. First, we design Analogical Memory Module (AMM), which discovers analogical exemplars for a given query scenario, from a reference pool of observed samples. Second, we propose Analogical Reasoning Networks (ARN), which takes the retrieved samples as input, selects candidate *analogy pairs* from the relevant reference samples, and learns proper reasoning operations over the selected analogy pairs, resulting in an *analogy context vector*. Third, we devise a Conditioned Composition Engine (CCE), which combines the analogy context vector with the representations of the query sample to predict the masked words and complete the target sentence with a novel composition.

We show how ARTNET generalizes to new compositions and excels in visually grounded language acquisition by designing experiments in various evaluations: novel composition prediction, assessment of affordance, and sensitivity to data scarcity. The results on the ego-centric video dataset (EPIC-Kitchens) demonstrate the effectiveness of the proposed solution in various aspects: accuracy, capability, robustness, etc. The project code is publicly available at https://github.com/XX.

The main contributions of this paper include the following:

- We call attention to a challenging problem, compositional generalization, in the context of machine language acquisition, which has seldom been studied.

- We propose ideas supported by human analogical reasoning: approximating new verb-noun compositions by learned arithmetic operations over relevant compositions seen before.

- We propose a novel reasoning-augmented architecture for visually grounded language acquisition, which addresses the compositional generalization problem through association and analogical reasoning.

- We evaluate the proposed model in various aspects, such as composition prediction, validity test, and robustness against data scarcity. The results show that ARTNET achieves significant performance improvements in terms of new composition accuracy, over a large-scale video dataset.

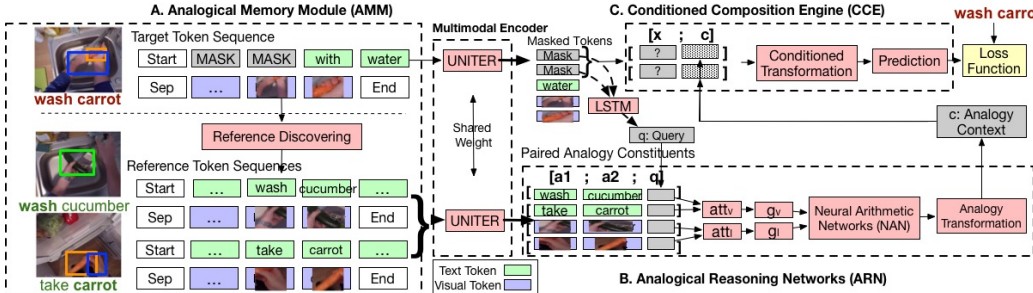

Figure 2: **Analogical Reasoning Transformer Networks (ARTNET):** The proposed reasoning-augmented architecture for compositional generalization via analogical reasoning.

## 2  ARTNET: ANALOGICAL REASONING TRANSFORMER NETWORKS

Our goal is to develop a framework that can support *multimodal compositional generalization* through learning in a visual-textual environment. The proposed framework learns to acquire the meaning of phrases and words from image-sentence pairs and to create novel compositions via reasoning. We call the framework Analogical Reasoning Transformer Networks (ARTNET), due to its ability to establish analogies with the previously seen, relevant scenarios, and perform reasoning operations to generalize a composition for the new scenario. Figure 2 illustrates an overview of ARTNET, which is composed of a multimodal encoder backbone, followed by three main modules: Analogical Memory Module (AMM), Analogical Reasoning Networks (ARN), and Conditioned Composition Engine (CCE). We elaborate each component in the rest of this section.

### 2.1  MULTIMODAL ENCODER BACKBONE

The backbone network is responsible for encoding image-sentence pairs into compositional semantic representations. To achieve this, we utilize the emerging multimodal transformers (*e.g.* UNITER (Chen et al. (2019)) or ViLBERT (Lu et al. (2019))), which have recently achieved great success in various vision-language tasks. These models take a set of visual and textual tokens (e.g. objects and words), and extract a multimodal embedding for each token, which is contextualized by all the other tokens through layers of multi-head attention. We follow the architecture of UNITER, as it performs slightly better than ViLBERT and other similar models. Note that since our goal is language acquisition, we intentionally do not use the pretrained weights of UNITER, which are trained on a large-scale corpus. Instead, we train the backbone from scratch on our limited data.

### 2.2  ANALOGICAL MEMORY MODULE

AMM plays the role of analogical association. Like finding a useful puzzle piece, we propose AMM to discover the most useful reference samples for analogical reasoning in a target scenario. Given a target image-sentence pair (query), where some tokens in the sentence are masked, we randomly select N (N = 200 in our experiments) sample image-sentence pairs from the training data to create a reference pool, and find the Top-K most relevant exemplars from that pool. To this end, we measure a multimodal relevance score between the query and each reference. Here, we use the initial embedding of each token on the query and reference samples as described in the Section 2.1. Given a target and a reference sample, we define the multimodal relevance score as a combination of visual and text relevance between the corresponding sets of tokens. For visual tokens, we compute the mean cosine similarity of every pair of tokens from the query and reference token sets. For the language part, the contextual background words that are not masked can provide linguistic clues for semantic relevance. Thus, we compute the Jaccard Index (Hamers et al. (1989)) between two sentences as textual relevance. Specifically, the multimodal relevance score is

$$s_{vl} = \frac{1}{2} \cdot \left( \frac{|W_T \cap W_R|}{|W_T \cup W_R|} + \frac{1 + \frac{\sum_i \sum_j cos(v_{T_i}, v_{R_j})}{N_v}}{2} \right) \tag{1}$$

where $W_T$ and $W_R$ are the set of target words and reference words, $N_v$ is the number of visual token pairs, and $v_{T_i}$ and $v_{R_j}$ represent the visual embeddings of the $i_{th}$ visual token of the query and the $j_{th}$ visual token of the reference. After computing the scores, AMM ranks reference samples with respect to their relevance scores and selects the Top-K most relevant samples for the given query.

## 2.3 ANALOGICAL REASONING NETWORKS

Given the retrieved analogical exemplars, we devise a neural network with reasoning ability to enrich the original representation of the masked compositions by making analogies with the seen compositions. The idea is to exploit the semantic relation mapping between the candidate analogy composition and the target composition. To this end, we represent the target masked composition as a query vector $q$, by concatenating the multimodal transformer embeddings of the masked words of that composition (typically a verb and a noun from the target sentence) and learning the representations of ordered constituents in a composition based on a Long Short-Term Memory (LSTM) (Zhou et al. (2015)). Next, we apply the multimodal encoder backbone (as mentioned above) on the retrieved analogy samples, and parse each sample into candidate analogy compositions (pairs of tokens). Since the goal is language acquisition, we do not rely on predefined grammar rules or pretrained models to generate the candidate compositions, such as applying part-of-speech tagging and taking each verb-noun pair. Instead, we enumerate all pairs of adjacent words from each retrieved sentence, and all pairs of detected image regions from each retrieved image. The multimodal resulting set of pairs are called analogy pairs hereafter.

The core of ARN consists of three neural network modules for analogy attention, analogical reasoning, and analogy transformation. Analogical attention learns the importance of each pair of candidate analogy composition and query vector respectively and generates analogy aggregation from each modality independently. Analogical reasoning is designed to learn the appropriate arithmetic operations from analogy compositions for reasoning. It consists of modality-wise transformations and Neural Arithmetic Logic Units (Trask et al. (2018)) with multiple layers of Neural Accumulator (NAC) (Trask et al. (2018)). NAC is a simple but effective operator that supports the ability to learn addition and subtraction. This module is applied on the analogy pairs, and computes a single vector that represents the output of some reasoning operations, optimized for our task through gradient descent. Through the analogy transformation, ARN generates the sequential representations of final analogy context vector. Specifically, ARN can be denoted as

$$c_a^m = \sum_j \alpha_{ij}^m h_j^m, \alpha_{ij} = \frac{\exp a([r_i^m; r_{i+1}^m; q], [r_j^m; r_{j+1}^m; q])}{\sum_k \exp a([r_i^m; r_{i+1}^m; q], [r_k^m; r_{k+1}^m; q])} \quad \text{Analogical Attention,} \quad (2)$$

$$h_c = f_{\text{NAC}}([g_v(c_a^v), g_l(c_a^l)]^T) \quad \text{Analogical Reasoning,} \quad (3)$$

$$c = \text{LSTM}(h_c) \quad \text{Analogy Transformation,} \quad (4)$$

where $v$ and $l$ represent the vision and language modalities, $r_i^m$ and $r_{i+1}^m$ ($m$ is modality indicator) are the image regions or text words of candidate analogical compositions. $g_v$ and $g_l$ are modality transformations that contains two-layer fully connected networks with ReLU activation and dropout, and $T$ represents matrix transpose. The output of ARN is the vector $c$, which is called analogical context vector, and will be used to augment the composition representations.

## 2.4 CONDITIONED COMPOSITION ENGINE

After analogical reasoning, we create a potentially novel composition based on both the initial comprehension of the given scenario and the result of analogical reasoning. To this end, CCE is designed to enrich the representations of the masked tokens through a conditioned learning network. It takes the analogical context vector as contextual knowledge. Let $x = \{x_1, ..., m_i, m_{i+1}, ..., x_N\}$ be the input elements of a multimodal encoder, and $m_i^l$ or $m_{i+1}^l$ are the $l$-th layer of the representations of the masked words. CCE uses the multimodal transformers to transform the embedding features of both each masked word and the analogical context vector. Then it predicts the masked word by aggregating from linguistic clues of all the other unmasked elements. The embedding feature $h_i$ of

the $i$-th masked word computed by:

$$h_i^{l+1} = W_2^{l+1} \cdot \text{ReLU}(W_1^{l+1}[m_i^l; c] + b_1^{l+1}) + b_2^{l+1} \qquad \text{, Context-conditioned} \qquad (5)$$

$$h_i^{l+2} = \text{GeLU}(W_i^{l+2} h_i^{l+1} + b_i^{l+2}) \qquad \text{, Feed-forward} \qquad (6)$$

$$h_i = \text{LayerNorm}(h_i^{l+2}) \qquad \text{, Feed-forward} \qquad (7)$$

where $W_1^{l+1}$, $W_2^{l+1}$, $b_1^{l+1}$ and $b_2^{l+1}$ are learnable weights and biases for the context-conditioned transformation. $W_i^{l+2}$ and $b_i^{l+2}$ are learnable weight and bias for feed-forward transformation, respectively. Given the contextual representation of the masked word $h_i$, the model predicts the masked word by multiplying its contextual representation with a word embedding matrix $\phi$ which is trained with the rest of the network, $\hat{w}_i = \phi_w^T h_i$.

## 2.5 LEARNING OBJECTIVES

**Masked Composition Acquisition** Our model learns to perform language acquisition by filling in masked words. At both train and test time, we give the model a target example, with a reference set sampled from the training dataset. Note that our masking policy is different at training time and test time (details in Section 3.1). During training and validation, we randomly mask multiple words and visual tokens. However, during test, we only mask one verb-noun composition.

**Objective Function** We train the model to acquire words by directly predicting them. We measure the Cross-Entropy loss between the predicted word $\hat{w}_i$ and true word $w_i$ over a vocabulary of size $C$, denoted as $\mathcal{L}_1$. This objective is the same as in the original BERT (Devlin et al. (2018)). We also learn visual reconstruction via a regression task. The visual loss $\mathcal{L}_v$ is a Triplet Metric loss (Weinberger et al. (2006)) to force a linear projection of $v_i$ to be closer to $\phi_v(v_i)$ than $\phi_v(v_{k\neq i})$, and $\phi_v(\cdot)$ is the visual representation network (ResNet). Because the objectives are complementary, we train the parameters of the proposed model by minimizing the sum of losses:

$$\min_{\Theta} \left( -\sum_i^C w_i \log(\hat{w}_i) + \lambda \max(0, m + \|v_i - \phi_v(v_i)\| - \|v_i - \phi_v(v_{k\neq i})\|) \right) \qquad (8)$$

where $\lambda \in \mathbb{R}$ is the parameter to balance the loss terms (modalities), $m$ is the triplet margin and $\Theta$ represents the trainable parameters of the entire network.

## 3 EXPERIMENTS

We compare our method (ARTNET) and baselines on new and seen composition acquisition tasks. To demonstrate the quantitative and qualitative results, we evaluate our method in a variety of aspects including performance comparison, validity test, incremental data setting, and case study.

## 3.1 EXPERIMENT SETTINGS

**Dataset** We use the instruction dataset EPIC-Kitchens (Damen et al. (2018)) in our experiments. The dataset consists of 55 hours of egocentric videos across 32 kitchens. Each video clip has a narrative sentence, so we create image-sentence pairs by selecting one key frame of each video clip. For each video frame, we use the object bounding boxes officially provided with EPIC-Kitchens, which are produced by running faster R-CNN (Ren et al. (2015)) too. We discard the object labels due to strict constraints in the language acquisition scenario. Importantly, we train our model our video dataset only, without any language pre-training. We partition the dataset to ensure new compositions used in testing have never been seen in training. The test subsets of new/seen composition tasks are disjoint (without overlap) and equal-size (each set contains 29K samples related to 238/861 unique new/seen compositions); the train subsets of the tasks are identical (142K samples with 3675 unique compositions). The dataset was prepared by the following steps: (1) We took all annotated compositions from the Epic-Kitchens and split them to two groups of compositions - seen consisting of 3675 unique compositions, 124 verbs and 254 nouns (vocabulary size), and new consisting of 238 new compositions, 90 verbs and 149 nouns. All the verbs or nouns of the new compositions are in the seen compositions too. (2) We randomly sampled 140K samples from the seen composition group as a shared train set and each seen composition has at least one sample in the train set. We

| Models / Metrics (%) | New Composition | | Seen Composition | |
|---|---|---|---|---|
| | Top-1 Acc. | Top-5 Acc. | Top-1 Acc. | Top-5 Acc. |
| BERT (from scratch) | 2.25 | 6.85 | 6.47 | 26.42 |
| BERT (pre-trained) | 2.06 | 10.41 | 7.89 | 27.10 |
| Multimodal BERT (language pretrained) | 3.05 | 23.25 | 14.79 | 35.05 |
| Multimodal BERT (from scratch) | 5.97 | 36.63 | **23.22** | **57.81** |
| **Proposed ARTNET** (from scratch) | **8.00 (+2.03)** | **41.08 (+4.45)** | 22.25 | 57.74 |

Table 1: **Performance Comparison:** Accuracy comparison on new and seen composition tasks.

also sampled a set of seen composition samples (about 20% of the train set) as the test set of seen compositions. (3) The samples of the new compositions were used to create the test set of the new compositions (size about 20% of the train set). To ensure the focus is on new compositions, rather than new words, we removed new compositions that contain new words not seen in the train set.

**Evaluation Metrics**   To compare the performance of our model against baselines, we evaluate the ability to acquire new or seen compositions (*e.g.* verb-noun word pairs). During testing, the model takes paired image-sentence data as input, and the target verb-noun composition is replaced with a special "mask" token. The objective is to predict the masked words from the complete vocabulary of all words (unaware of the part of speech). We adopt Top-1 and Top-5 accuracy to measure the performance, which calculates the average accuracy over the predicted nouns and verbs with the top N highest probability. The prediction is correct when both noun and verb are correctly predicted.

## 3.2   PERFORMANCE COMPARISON WITH BASELINES

We compare our model with state-of-the-art methods of language modeling or multimodal language modeling (implementation and more training details of our model and baselins are in the Section 6 Appendix). We consider three variants of the BERT family (Devlin et al. (2018); Chen et al. (2019)):

**BERT (language only)** is a powerful language transformer model (Devlin et al. (2018)) that achieved state-of-the-art performance in multiple tasks. We adopt two learning settings: (1) train the BERT model from scratch on the sentences of our task data; (2) utilize the pre-trained BERT model (Zhu et al. (2015); Chelba et al. (2013)) and fine-tune BERT on the task data. Note that the pretrained BERT has the advantage of large-scale data, which is different with language acquisition.

**Multimodal BERT (language + vision)** is a generalization of BERT that receives both visual and textual tokens as input, which usually come from an image-sentence pair. Here we use the same architecture as our backbone UNITER (Chen et al. (2019)), and we adopt a pre-trained ResNet-18 (He et al. (2016)) to extract visual features. We train it from scratch on our task dataset.

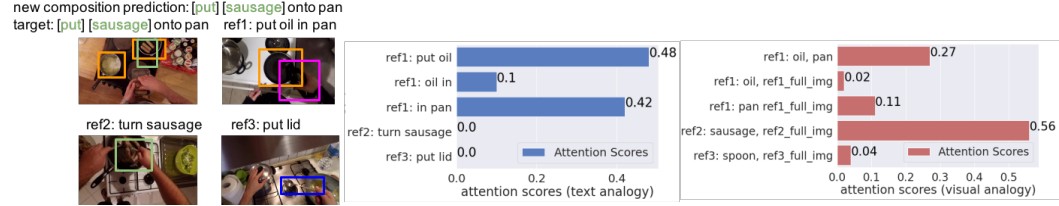

Figure 3: **Reasoning Attention Visualization over Analogy Pairs:** The bar charts shows attention scores in our reasoning module for textual (left chart) or visual (right chart) analogy pairs.

The quantitative and qualitative results are illustrated in Table 1 and Figure 7 (in the appendix section), respectively. For new composition prediction, as shown in Table 1, our model can significantly achieve 2.03% and 4.45% improvement over state-of-the-art baseline on Top-1 and Top-5 accuracy respectively. Such improvement results from the analogical reasoning based on relevant reference samples. For seen composition prediction, the proposed model is nearly unchanged. To understand what the analogical reasoning module has learned, we visualize the attention distribution over multimodal analogy pairs in Figure 3 (detailed analysis and more examples are in Figure 8 and para-

| Composition Acquisition | Affordance Accuracy | | | Table 2: **Validity Test:** Performance comparison of validity accuracy. |
|---|---|---|---|---|
| Methods/Settings | Overall | New Comp | Seen Comp | |
| BERT (w/o vision) | 81.67% | 81.59% | 81.72% | |
| Multimodal BERT | 85.79% | 84.78% | 86.36% | |
| **Proposed ARTNET** | **86.48**% | **86.01**% | **86.75**% | |

graphs in the appendix section). As shown in Figure 7, the model can successfully retrieve relevant reference samples that contain analogical pairs (detailed analysis is in the appendix section).

### 3.3 EXPERIMENTS ON DIFFERENT TRAINING SIZES

To evaluate our model on low-data regime and simulate the language acquisition process of a child, we train on different scales of data. Specifically, we consider 5 different training data size percentages (TPs) (100%, 80%, 60%, 40% and 20%), and plot the performance of our method compared to the stringest baseline in Figure 4. The ARTNET achieves 13.44% Top-5 accuracy with only 20% of training data, which has larger gap with the baseline, suggesting our stronger generalization ability.

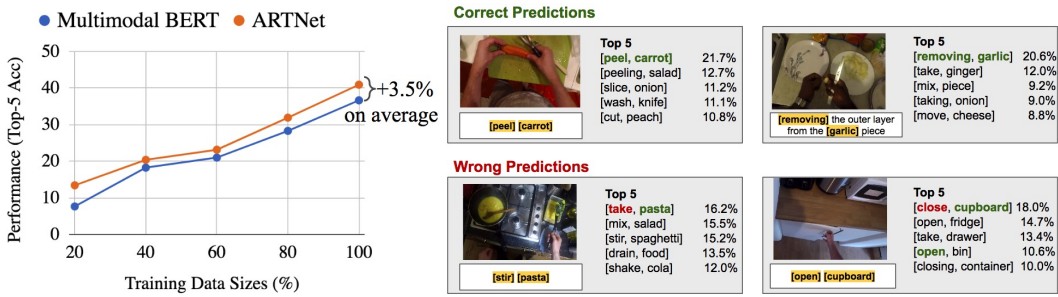

Figure 4: **Robustness:** Performance curve with different sizes of training data.

Figure 5: **Case Study:** Top-5 prediction results for new compositions (highlight words are masked ground-truth compositions to be predicted).

### 3.4 VALIDITY TEST EVALUATION

We propose a way to aim at assessing whether the predicted results follow human commonsense in the real world, called "Validity Test", based on the definition of affordance (Gibson (2014)). Affordance is a binary relationship between each noun and verb, which indicates whether a certain verb is typical to involve a certain noun as an argument (Baber (2018)). For examples, "cutting pizza" and "cutting carrot" happen in our daily life, but we never see "cutting plate" or "cutting water". We annotate the validity accuracy for 8400 Top-1 verb-noun compositions (5704 of which appear in our dataset) predicted by each model given any test image. The type-mismatched compositions, such as "meat apple" (noun-noun) or "peel cut" (verb-verb), are also defined as invalid prediction. In Table 2, we show the Top-1 validity accuracy (the ratio of the number of affordable or valid predictions to all) for each model. Overall, ARTNET achieves higher validity than our baselines 86.5%. The validity of our new composition results outperforms Multimodal BERT/BERT with 1.23%/4.42% improvements, which is significant. We attribute such gains in composition validity to the novel components in our model - discovery of multimodal analogy reference samples and reasoning over such samples to generate the language description. The method's emphasis on analogical relations to realistic reference samples helps improving the validity of the compositions predicted by the model.

### 3.5 CASE STUDY

To analyze the strengths and failure cases, we select some examples shown in Figure 5. Intuitively, vague visual clues such as ambiguous action and tiny objects, are hard to recognize. However, we observe ARTNET has a unique ability to disambiguate such difficult cases, as shown in the top row of Figure 5. Nevertheless, the model fails in some cases, such as take/stir and close/open. This is mainly due to the fact that we used still keyframes rather than dynamic videos, which discards

Figure 6: **Case Study:** The validity test results. The two words in brackets are masked and to be predicted as the target composition. Green or red color indicates valid or invalid prediction.

motion information. Future work should incorporate spatio-temporal features to alleviate this limitation. Moreover, we provide several cases with Top-5 prediction results to show the affordance and rationality of our method's prediction. Considering Figure 6, we observe that not only the Top-1 results, but also most of the Top-5 are in line with human commonsense, and superior to the baseline.

## 4 RELATED WORK

**Compositional Generalization** Compositional generalization is an important open reasoning challenge (Keysers et al. (2019); Chang et al. (2018); Loula et al. (2018)) and a priority to achieve human-like abilities (Bastings et al. (2018)), regarding the ability to compose simple constituents into complex structures (Nikolaus et al. (2019)). From the traditional view, numerous studies utilize linguistic principle to explore compositionality (Mitchell & Lapata (2008); Baroni & Zamparelli (2010)). With recent advances, there are more attempts to solve compositional generalization with neural networks for the tasks with synthetic commends or interactions (Lake (2019); Russin et al. (2019); Li et al. (2019); Kato et al. (2018)). The current machine intelligence still lacks compositional generalization ability, because they are prone to fail on such tests in realistic or natural scenarios (Nikolaus et al. (2019); Loula et al. (2018); Keysers et al. (2019)). Moreover, several recent works also try to enhance compositional generalization for natural language (Nikolaus et al. (2019)) with pretrained representations or lexical knowledge. But there are few works addressing this emerging and valuable challenge for language acquisition in a multimodal reasoning view.

**Visually Grounded Language Acquisition (VLA)** Like the language acquisition of kids, VLA is a task of acquiring language constituents from scratch within a visual-textual environment. Although the works of grounded language learning achieve the good progress in many tasks (such as visual captioning (Yu & Siskind (2013); Kiela et al. (2017); Ross et al. (2018)) or robotics (Matuszek (2018))), there are major differences between VLA and multimodal pretraining, including the goal of learned model, the data amount needed, and the the architecture condition (e.g., whether use the predefined language parsing). The former attempts to learn from scratch aspects of language models (e.g. compositional semantics) and effectively use limited samples, while the latter uses context from very large training corpora (Lu et al. (2019); Chen et al. (2019)) to learn language representation by a data-hungry model. Several recent works further address acquiring concrete language including word representation (Kottur et al. (2016); Surís et al. (2019)), compositional semantics (Jin et al. (2020)) from scratch via visual clues, which are more related to our task. Our work seeks to improve, in the form of reasoning, the constituent generalization to new compositions.

## 5 CONCLUSION

In this paper, we take a step towards visually grounded language acquisition, by studying the problem of compositional generalization in the state-of-the-art multimedia language models. Inspired by the human brain's analogical reasoning process, we propose to form new compositions by recalling observed compositions and relating them to the new scenario through learned arithmetic operations. Our proposed reasoning-augmented method, Analogical Reasoning Transformer Networks, achieves superior compositional generalization capability compared to the state-of-the-art transformers, which results in significant and stable performance gains in unseen compositions over a large-scale instructional video dataset.

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

## 6 APPENDIX

**Implementation Details**  In all experiments, training is conducted on 2 GPUs for 200 epochs. In each mini-batch, 256 samples are drawn for each GPU and in each sample, image regions are cropped from the whole image and resized to $112 \times 112$. The transformer encoder in all models has the same configuration: 4 layers, a hidden size of 384, and 4 self-attention heads in each layer. Besides, we use AdamW optimizer  Loshchilov & Hutter (2017) with the learning rate 3e-5. For AdamW optimizer  Loshchilov & Hutter (2017), we set the update coefficients, for averages of gradient and its square ($\beta_1$, $\beta_2$), and $\epsilon$ on denominator as 0.9, 0.999, 1e-4. During training, we mask out text tokens $\frac{1}{3}$ of the time and image tokens $\frac{1}{6}$ of the time and follow the same setting of random masking strategy with BERT  Devlin et al. (2018).

During training and testing, for each target sample, we randomly select 200 remaining samples as the corresponding reference set. In our experiments, we utilize Top-K (K = 3) reference samples to get involved in analogical reasoning.

In evaluation, we calculate the accuracy based on the same random mask strategy as training process. An early stop strategy is utilized based on the Top-5 Acc. in validation that the training process will terminate if the validation Top-5 Acc. doesn't increase again.

**Target-Reference Case Study**  From the two examples shown in Figure 7, it's easy to observe that the model can successfully retrieve relevant reference samples which contain analogical pairs, by computing their visual and language similarity to the target composition. For the correct prediction "peel carrot", the model discovers "stir carrot", "peel potato" and "cut potato" for analogical reasoning, while for "wash knife", the model retrieves "wash plate", "take knife" and "rinse knife" as reference samples.

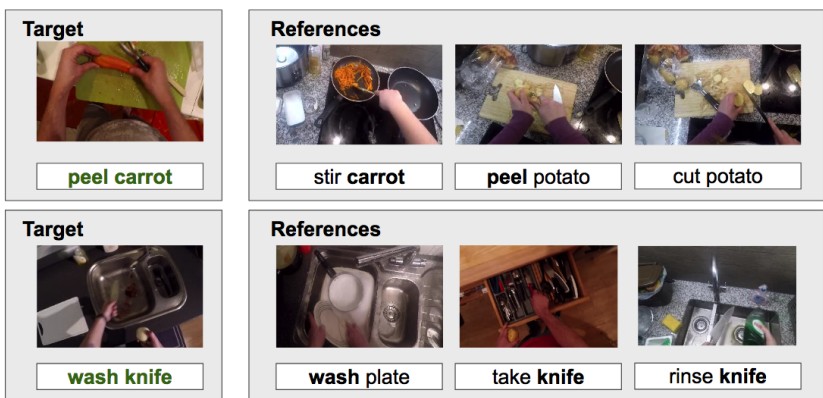

Figure 7: **Case Study:** The target sample and top samples discovered from the reference set.

**Reasoning Attention Distribution over Multimodal Analogy Pairs**  In the four examples shown in Figure 7, we provide two examples for correct prediction and another two for wrong predictions. From the bottom two examples, the model learns compositional semantics from both visual and textual constituents. For the correct prediction of new composition "put sausage", the model learns to acquire and approximate novel composition from our multimodal reasoning. The reasoning has more attention on the textual phrase of the first reference sample "put oil in pan" and visual regions "sausage, whole image" of the second reference sample. This implies that the model learns textual and visual semantics from reference samples and compose them under similar scenarios as context. A similar phenomenon also appears in the second prediction example for seen composition "chop onions". The model is able to learn the phrase "chop onion" from different modalities. For wrong prediction results, the minor visual differences of several verbs will lead to wrong reasoning (e.g. "remove skin of garlic"), although the model can successfully retrieve relevant reference samples with aid by contextual information. While chopped garlic is not visually recognized by the adopted vision model, the attention distribution of the visual analogy pairs for the example seems to focus on the garlic but also be confused by "cream, whole image". Meanwhile, the accuracy of the reasoning

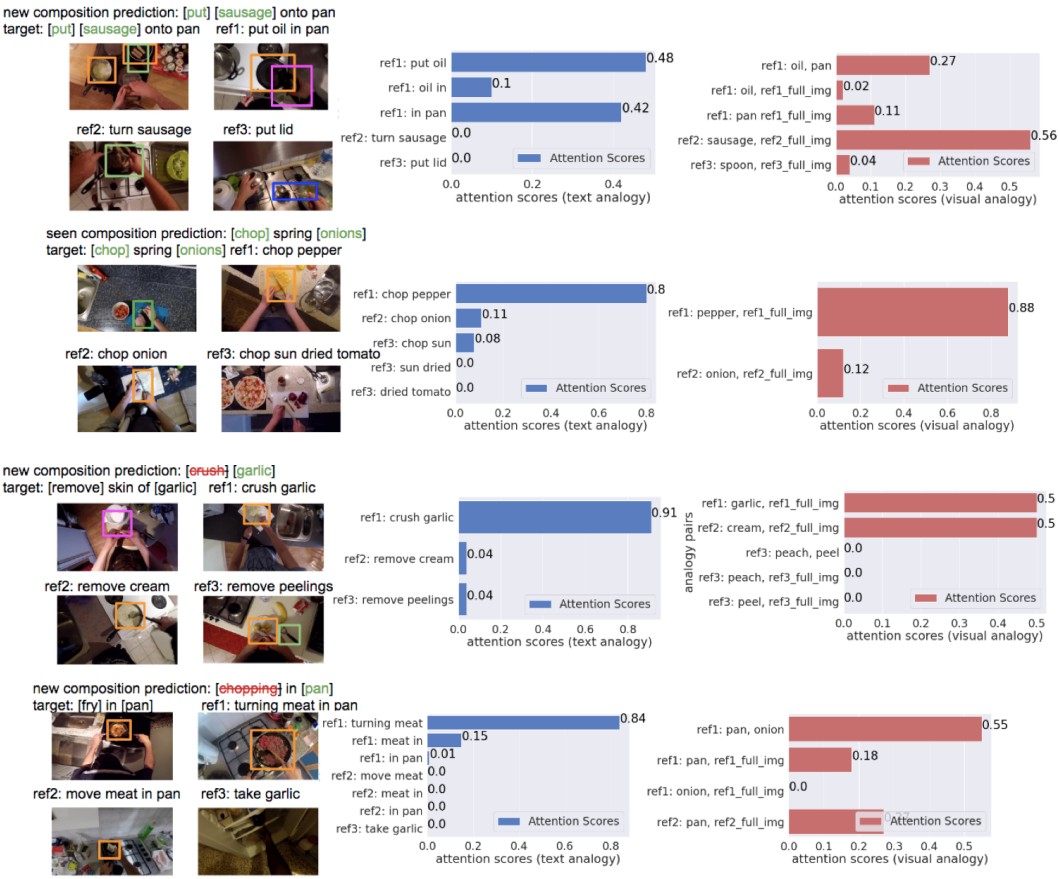

Figure 8: **Reasoning Attention Visualization over Multimodal Analogy Pairs (Correct or Wrong Predictions):** The bar charts shows attention scores in our reasoning module for textual (left chart) or visual (right chart) analogy pairs.

is also impacted by the relevant samples (e.g., "fry in pan"). When the model didn't discover "fry" in the relevant references and can not distinguish the actions ("fry" and "chop") in a target and a reference sample, the reasoning would easier to get the wrong prediction.

