# OpenReview forum: "Analogical Reasoning for Visually Grounded Compositional Generalization"
_ICLR.cc/2021/Conference — Reject_

### Official Review · AnonReviewer1 · 2020-10-22
**A straightforward task with a complex architectural approach, unclear contribution scope, and missing baselines.**

**Rating:** 3
**Confidence:** 3

**Review:**

This paper presents a model that takes in a keyframe from a video and emits the noun and verb best matching what is being done in the frame.
At inference time, the noun and verb have never been seen in combination with one another, but have each been seen paired with other nouns/verbs at training time.
The paper presents a complex, three part model (ArtNet) to tackle this challenge, as well as unimodal linguistic baselines.
Notably, the evaluation does not include vision-only baselines or pretrained model toplines, making it difficult to assess exactly what ArtNet is learning and where its advantage lies.

Questions:

- There are multiple references to "creation", e.g., "create novel compositions" which makes the method sound like it's performing generation. From my understanding, though, ArtNet is purely discriminative, taking in a keyframe and predicting a noun and verb. However, there is one line in the paper that says "We also learn visual reconstruction via a regression task.", which makes it sound like there's a formulation of ArtNet that maybe takes in a noun and verb and produces a keyframe image (using a GAN, maybe? Or a nearest neighbor lookup?), and so does "create novel compositions". If that's the case, it isn't described, and this image reconstruction task is never mentioned again in the paper or described in any equations.

- Were pretrained ViLBERT/UNITER run alone as a topline? Establishing how much is lost in performance due to lack of pretraining + how this method addresses that with sparse data would make a much stronger argument, I feel. In particular, we would want to see pretrained ViLBERT/UNITER and then pretrained + ArtNET to give a sense of how performance will change as models have seen huge amounts of aligned data.
- [Related] In Table 1 what's the intuition for language pretrained mBERT/UNITER/ViLBERT falling behind from scratch? Why use language pretraining and not vision pretraining (e.g., topline)?


- In Eq (1), why use cosine similarity for visual embeddings but then back off to surface forms for words? Was cosine similarity for word representations computed by UNITER tried? What is the intuition for this not working, if it did not?

- Not sure in Eq (4) what the sequence input to the LSTM is; does c range over some sequence? Doesn't the sum already collapse that?

- "To ensure the focus is on new compositions, rather than new words, we removed new compositions that contain new words not seen in the train set." All new words or just nouns/verbs? It's a big advantage/relaxation on the test set to have no OOV tokens.

- Why no vision-only baseline? Strip word contexts at training time except noun/verb, then predict only noun/verb at test. A lot of this could be basically object recognition followed by activity recognition or strong priors on p(activity | object) (e.g., always "open" or "close" for cabinets).

- "outperforms Multimodal BERT/BERT with 1.23%/4.42% improvements, which is significant"; what statistical significance test was used? How many random initializations were tried to establish the average performance numbers for comparison between performance populations?

Suggestions for Improvement:

- "We call attention to a challenging problem, compositional generalization, in the context of machine language acquisition, which has seldom been studied." This is poorly worded, since compositional generalization is well and commonly studied, to the point that even in this paper there is a section in the related work about it. Major workshops also list compositionality as a topic of interest, so I don't think it's fair at all to say that this has "seldom been studied" [ https://sites.google.com/view/repl4nlp2020/home ]. In the context of language acquisition specifically, emergent communication work focuses heavily on composition [ https://sites.google.com/view/emecom2019/home ].

- ViLBERT in intro, UNITER in description of method, "Multimodal BERT" (mBERT?) in Table 1. What was used? Needs to be consistent in presentation.

- "We discard the object labels due to strict constraints in the language acquisition scenario." Because Faster RCNN is trained on ImageNet, which is based on WordNet, the object categories still exist in the form of supervision. The model has a linguistically-motivated notion of what an "object" is that can be traced to the WordNet. This should be acknowledged; you can't actually "get rid" of linguistic information inherent in a pretrained Faster RCNN.

- "But there are few works addressing this emerging and valuable challenge for language acquisition in a multimodal reasoning view." This paper does not really tackle language acquisition, though? There's a restriction so that the test set has no OOV words, even. I think the claims and presentation of the paper need to be carefully re-scoped.

- There is a lot of focus on "learned arithmetic operations" but no analysis as to what exactly this component ends up doing or learning.

Nits:
- Typo Introduction "a language model that generate" S/V agreement.
- Figure 1 doesn't feel like it communicates anything about the method, and does not seem tied to the caption. Are boxes (1, 2, 3) meant to represent the association, reasoning, and inference steps? What's happening in each?
- "The results show that ARTNET achieves significant performance improvements in terms of new composition accuracy, over a large-scale
video dataset." strange wording makes it sound like ARTNet is outperforming a dataset, not a method.
- "We train the model to acquire words by directly predicting them." this sounds like the model will be predicting words unseen at training time, which is not so. In particular, "acquire" here sounds like the model will be exposed to the word at most once (at inference time) and then be able to memorize that exposure in sequence.
- Typo? In 3.1 "by running faster R-CNN too" what is the "too" pointing to? Do you run Faster-RCNN somewhere?
- Typo 3.3 "stringest baseline" strictest?
- Figures 4 and 5 are so close together their captions bleed together and are really difficult to disentangle.
- Typo "than our baselines 86.5" makes it sound like the baselines achieved 86.5.
- Typo 4 "the goal of learned model" missing "the"

---

> ### Author Response · Authors · 2020-11-23
> **For AnonReviewer1 (Q1-Q8), Thank you for your review**
>
> We appreciate your comments about experiments and descriptions. We address all of your questions and concerns below.
> We added the two new results for references: a vision-only transformer and a Multimodal BERT (vision-language pre-trained, or vl pre-trained).
>
> **Table [new]**
>
> |Evaluation |new |composition|seen |composition|
> |---|---|---|---|---|
> |Accuracy(%)|top1|top5|top1|top5|
> |Vision-only Transformer (from scratch)  |2.60|13.09|9.55 |16.78|
> |Multimodal BERT(vl pre-trained)  |2.61|6.66  |3.42 |7.13  |
> |Proposed ARTNet|8.00|41.08|22.25|57.74|
>
>
> **Questions**
> * **Q1:**
>   * We changed the term “creation” or “create” to “generalize”  [1] to avoid confusion. It means generalization to new compositions that haven't been seen or included in the training data during language acquisition.
>   * We changed “visual reconstruction via a regression task” to “visual reconstruction via a regression loss on masked regions. The model is trained to regress the visual feature (not image) corresponding to each masked region.” We refer to this as visual reconstruction. We revised the description of the visual reconstruction loss (refer to [2]) in Section 2.5. During training, the multimodal objective function of ArtNet includes two loss components: masked image region reconstruction loss and masked word modeling loss. During testing, the task is masked composition modeling without visual feature regression.
>
> * **Q2:**
> In fact, using pre-trained Multimodal BERTs alone as a topline is not fair and different from the problem being addressed in this paper.
>   * It breaks the problem setting for novel composition generalization. 1) In our multimodal composition generalization task, new compositions should not have been seen during training. Using pre-trained Multimodal BERT (or pre-trained + ArtNET) means the incorporation of external multimodal data (e.g., MS-COCO) which actually includes new compositions that we want to test. 2) For comparison, we have already used a Multimodal BERT (trained from scratch) in Table 1 as one of the strong baselines.
>   * One important goal of our research is the generalization of novel composition using multimodal language acquisition. This is different from the challenge of data sparsity.  The dataset we used includes 142K training samples. So strictly speaking it’s not really sparse.
>   * To investigate the effect of data size, we also provided the results over different data sizes (20%-100% of the full set) in Section 3.3.
>
> * **Q3:**
>   * Adding language pre-training to Multimodal BERTs may further improve the representation of input contextual words but do not be adopted on masked words or solve multimodal compositional generalization. Due to the domain difference of the datasets (between the pretraining domain and the target domain), we do not expect the pretraining to be significantly better than the BERT trained from scratch. This can be seen in comparing BERT (from scratch) with BERT (pre-trained) also - very minor performance gain (sometimes even performance drop) was seen.
>   * The reason for using language pretraining is to demonstrate the task cannot be solved well even having the representations of all single words (pre-trained language embeddings). And the studied challenge is composition generalization (for language composition acquisition), which is formulated as a weakly-supervised masked composition modeling. To avoid leaking of the fully-supervised image region to word grounding information, we do not use vision pretraining (for detected visual regions).
>   * We added a vision-related pretraining model in Table [new]: a pre-trained Multimodal BERT ( weakly-supervised vision-language pretraining).
>
> * **Q4:**
>   * For language composition acquisition, our model attempts to learn the masked language composition representations from weakly-supervised multimodal data. Although it allows visual embeddings for the image region representation, only one-hot vectors for the word representation as input to avoid pre-trained word embedding (as external word representation). So we use another way for words instead of cosine similarity.
>
> * **Q5:**
>   * The LSTM module takes a sequence of output vectors $[h_0,..., h_c, …, h_N]$ as input from the reasoning module Eq(3) where $c$ ranges in $N$ analogy pairs. We revised $h_c$ in Eq(4) to $[h_0,..., h_c, …, h_N]$ ($c \in [0, N]$) and added detailed descriptions.
>
> * **Q6:**
>   * Only nouns/verbs. We changed the “contain new words not seen in the train set” to “contain new noun/verb words not seen in the compositions of train set” to clarify.
>
> * **Q7:**
>   * In fact, object recognition or prior is not the main reason for the performance gain of the proposed model. As shown in "Table [new]", the vision-only baseline performances are significantly lower than the ARTNet model.
>
> * **Q8:**[corrected]
>   * We changed the description to “with significant improvements on the validity accuracy (1.23%/4.42%).”
>   * We test 5 times for each model.

---

> > ### Author Response · Authors · 2020-11-23
> > **For AnonReviewer1 (S1-S5, N1-N9), Thank you for your review**
> >
> > **Suggestions for Improvement**
> > * **S1:**
> >   * We removed the description “has seldom been studied”: “We aimed at the multimodal composition generalization for machine language acquisition. The difficulties of our task come from both weakly-supervised compositional generalization and language composition acquisition from scratch without external language representations.” We also added those references in the related work.
> >
> > * **S2:**
> >   * We clarified the description “We reimplemented a multimodal transformer (called Multimodal BERT), the architecture of the transformer layer is similar to a single stream multimodal BERT (UNITER [2])”
> >
> > * **S3:**
> >   * In fact, we used the Faster RCNN for feature extraction without word representation labels. The model only knows which image regions visually similar by visual features, and category labels do not leak linguistic information. So the model does not use the word labels of objects.
> >
> > * **S4:**
> >   * We changed “challenge for language acquisition" to “compositional generalization challenge in language acquisition”.
> >   * In fact, the paper studied the compositional generalization challenge in language acquisition. The model did acquire composition knowledge and distributed word vectors as representations (embedding vectors) from scratch. And each input word token sequence has only a set of word symbols (one-hot vectors) without word representations or composition semantics.
> >
> > * **S5:**
> >   * We provided the Fig. 7 Case Analysis and Fig. 8 Reasoning Attention Visualization in the append section. Both visualizations demonstrate the learned results and attention streams during learning.
> >
> > **Nits**
> > * **N1, 7-9:** Thank you. We corrected all the typos, description, and layout that you mentioned.
> > * **N2:** We use it to show a diagram of our ideas. Yes, the three parts are an association, reasoning, and inference. The detailed descriptions in each step are in the three boxes of Fig. 1 and are corresponded with the three stages  (A, B, and C) of our framework architecture in Fig. 2.
> > * **N3:** We changed it to "The results show that ARTNET achieves significant performance improvements in terms of new composition accuracy, on a large-scale video dataset."
> > * **N4:** We changed “model to acquire words by directly predicting them” to “model to learn new compositions by directly predicting them”. Because the model will be exposed to the new compositions at most once (at inference time).
> > * **N5 and N6:** Yes. “stringest” should be “strongest”
> >
> > References:
> > * [1] Brenden M Lake. Compositional generalization through meta sequence-to-sequence learning. NeurIPS, pp. 9788–9798, 2019.
> > * [2] Chen, Yen-Chun, et al. "Uniter: Learning universal image-text representations." arXiv preprint arXiv:1909.11740 (2019).

---

> > ### Comment · AnonReviewer1 · 2020-11-24
> > **Response**
> >
> > [Table New] The vision-only transformer results are good to see and fit intuition. I'm not sure I follow what was done for the "Multimodal BERT(vl pre-trained)". It seems it's outperformed by both "Multimodal BERT (from scratch) " and "Multimodal BERT (language pretrained)". Was it just run cold without any fine-tuning on this data? That would be an inappropriate comparison, since the (from scratch) and (langauge pretrained) models get fine-tuned on this task training data. If it was fine-tuned, what's the intuition for it performing worse than being trained from scratch? Is this task so radically different from existing pretraining data that there's no lang/vis alignment in previous tasks that generalizes?
> >
> > [Q1] Thanks for this clarification. In that case, the loss looks a lot like UNITER / ViLBERT. Maybe it would be easier to make that comparison explicit?
> >
> > [Q2/Q3] This makes sense. Maybe mention explicitly in the paper why such toplines actually "cheat" given the task formulation just to remind the reader.
> >
> > [Q5/etc.] Can you upload a revised version of the paper to reflect these claimed changes? The last revision is still in October in the system.
> >
> > [Q8] If still using the word "significant", this claim needs to be backed up with a statistical test. With 5 runs per model, you might have enough power to run a students t. It isn't clear that the models are being run multiple times currently since there are no standard deviations reported.

---

> > > ### Author Response · Authors · 2020-11-25
> > > **For AnonReviewer1 (Response), thank you for your response**
> > >
> > > * **[Table New]:**
> > >   * To clarify, when performing "Multimodal BERT (vision-language pre-trained, or called vl pre-trained)" in “Table [New]", we do not use fine-tuning. As explained in Q2and Q3, the Multimodal BERT (vl pre-trained)  is cheating if we use both other vision-language pre-training data (e.g. MS-COCO) and fine-tuning. So this one does not include fine-tuning.
> > >   * In comparison experiments of our paper (Table 1), the reason that we do not use pre-training by fine-tuning was explained in Q2 and Q3. Because such vision-language pre-training may have seen the new compositions (conflict to our compositional setting) in the pre-training data. Therefore is not compatible with the task for the task formulation.
> > > * **[Q8]:**
> > >   * We reported the overall standard deviations are $\pm$0.3% in the response to AnonReviewer4 - C2.1. And we added other deviations in the revised paper.
> > >   * T-tests for the validity accuracy improvements (1.23%/4.42%) on new composition in Section 3.4: We changed the description to “outperforms Multimodal BERT/BERT with significant improvements 1.23% (t=54.57, p=1.41e-11) and 4.42% (t=13.20, p=1.03e-06) on the validity accuracy.”
> > > * **[Q1], [Q2/Q3]:** OK.
> > > * **[Q5/etc.]:** Sure, I will upload a new version later if authors are allowed to add a new version of the paper during the discussion period.

---

### Official Review · AnonReviewer4 · 2020-10-28
**Clarity and ablation**

**Rating:** 5
**Confidence:** 4

**Review:**

Summary

The paper describes a model to predict unseen verb-noun compositions from a fixed set of verbs and nouns, given a training set of seen compositions. For an image with incomplete sentence, the model learns to select a set of relevant examples from the train set to serve as reference pairs of image-sentence mapping, and then uses these pairs to draw distribution over potentially unseen verb-noun pairs.


Strengths
- The problem is well-motivated
- The model is shown to perform well both quantitatively and qualitatively, along with different experimental setups such as validity, low-data


Weaknesses and concerns

- Hanging notations and unexplained variables:  Unfortunately this is a bottleneck for me to understand the paper thoroughly
    - Equation 2
        - What is $a$?
            - $a$ is used as a function within the softmax expression - $\exp a()$, but $a$ is also used as a subscript $c_a$, and since I cannot find any reference for $a$, I’m unclear as to what it actually is
        - The leftmost expression marginalizes out $j$ to yield $c_a^m$, but where does $i$ go? Are $a$ and $i$ related?
        - What is $k$? Is $k$ the cardinality of the set of all analogy pairs?
    - Equation 8 - Triplet loss
        - What is $v_i$?
- Results:
    - Potential non-determinism
        - You mentioned that for each test sample, you randomly pick 200 examples and then select top-K examples from them (ref: Section 2.2). Now if I understood this correctly, this introduces non-determinism in your results. Is it fair to say that your results should report standard deviations along with the numbers currently stated? Are the improvements statistically significant then?
    - Lack of ablation
        - Frequency bias in predictions? What’s the accuracy when you pick the most common verb, noun, and verb-noun pair?
        - Choice of K in Top-K
            - The Appendix (Implementation details) mentions that you pick K as 3. Why? What happens when you increase or decrease the value. Is there any insight to be gained from here?
        - Accuracy numbers for nouns and verbs separately
            - Again, this is an attempt to uncover biases in your predictions. You mark a prediction correct if both verb-noun pair gets correctly classified, but which one of them gets more wrong? Is there something to be learned here?
    - Trends in Table 1
        - BERT pre-trained v/s scratch: Why is there a reverse trend in top-1 and top-5 accuracy? Is there a conclusion to be drawn here?

- Misc
    - Number of objects (visual tokens) and words (lingual tokens) in an image?
        - What is the distribution of number of object and words in the dataset for an image-sentence pair?
    - Figure 3 - Attention over visual
        - What is “ref1_full_img”, “ref2_full_img”? Are complete images also included in analogy pairs along with objects?
    - Section 3.4
        - “We annotate the validity accuracy for 8400 Top-1 verb-noun compositions” How many people were involved? What does the distribution look like?
    - Reasoning module:
        - (e.g. “washing carrot” = “washing apple’ + “cutting carrot” - “cutting apple”)
            - Is this arithmetic observed in predictions?


Minor concerns (suggestions, typos, etc.)
- Section 3.4: grammatical errors


Preliminary Rating and its justification

The current version of the paper seems to have a couple of loose ends in terms of clarity and completeness of experiments. However, I’m open to changing my rating if the above concerns are addressed to some extent.

---

> ### Author Response · Authors · 2020-11-23
> **For AnonReviewer4 (C1-C2): Thank you for your review**
>
> We appreciate your comments in recognizing several of the novel ideas and thorough experiments. We address all of your concerns and questions below.
>
> * **C1- Notations:** We clarified and revised the mentioned notations and variables in Equation 2 and 8.
>   * **C1.1 - Equation 2 Analogical Attention**
>     * About “$a$”: “$a()$” function means the alignment functions for the attention mechanism [1]. The “$a$” subscript in $c^m_a$ means “attention” instead of a variable, and we changed $c^m_a$ to $c^m_{attn}$.
>     * About “$i$”: It indicates the index of the first masked word in each target sample. Each sample would correspond to an individual “$i$” value depending on the position of the first masked word. For the “$a$” subscript, please refer to the above answer: About “$a$”.
>     * About “$k$”: Yes. It is a cardinality which means the numbers of all analogy pairs.
>   * **C1.2 - Equation 8 Triplet Visual Loss**
>     * To clarify, we changed $v_i$ to $\hat{v}_i$ in the triplet visual loss. It means a representation vector (by our model) of a reconstructed (or generated) image region for visual self-supervision. The visual loss measures the loss between a reconstructed (or generated) representation $\hat{v}_i$ and a ground-truth representation $v_i$ of the image region of a training sample.
>
> * **C2 - Results:** We added all the suggested ablation experiments and descriptions.
>   * **C2.1 - Potential non-determinism**
>     * To avoid non-determinism and keep fairness, we use the same random seed in the random picking process for different ARTNet models.  Moreover, we performed 5 runs with different random seeds, the max standard deviation of the performance of our model $\pm$0.3%, relatively quite small compared to the large performance gain (more than 2% gain for top 1 accuracy). We added all standard deviations in the revised paper. For analysis of selecting top-K reference samples, please refer to the answer of “ C2.2 - Lack of ablation”.
>
>   * **C2.2 - Lack of ablation**
>     * **Frequency bias in predictions:** We added the ablation for frequency bias below. Our “frequent mode baselines” predict the most common verb, noun, or verb-noun pair as an answer for each sample. Using frequent words reaches very low accuracy:
>     |Evaluation      |new |composition|seen |composition|
>     |---|---|---|---|---|
>     |Accuracy(%)	|top1|top5|top1|top5|
>     |Popular verb-noun |0.41|1.28|0.78	|1.41|
>     |Popular verb 	|0.30|1.27|0.84|1.52|
>     |Popular noun 	|0.37|1.18|0.81|1.34|
>
>     * **Choice of K in Top-K:** We added the performances of the hyperparameter K. Increase or decrease of the K value will lead to minor performance changes. For each target sample, the ablation test shows too many or few reference samples are not helpful.
>     |Evaluation  |new |composition|seen |composition|
>     |---|---|---|---|---|
>     |Accuracy(%)|top1|top5|top1|top5|
>     |K=2		 |7.06	|40.61	|22.8	|56.83|
>     |K=3		 |8.00	|41.08	|22.25	|57.74|
>     |K=4		 |7.28	|41.23	|22.81	|57.21|
>
>     * **Accuracy for nouns and verbs separately:** We added the experiment. Performances for nouns are slightly higher than verbs, which shows verb representations could be further improved (e.g., incorporating motion vectors).
>     |Evaluation  |new |composition|seen |composition|
>     |---|---|---|---|---|
>     |Accuracy(%)|top1|top5|top1|top5|
>     |verb	  |9.23|43.45|23.04|59.82|
>     |noun  |10.74|45.77|24.23|62.25|
>
>   * **C2.3 Trends in Table 1**
>     * Overall, the performances of the pre-trained BERT (with language pretraining) are better than BERT (from scratch). Because using pre-trained word representations (from external large-scale language data) provides prior knowledge about language. But due to the domain difference of the datasets (between the pretraining data domain and our data domain) and the composition generalization difficulty, it’s reasonable that pretraining does not significantly boost the performances for new compositions (sometimes even worse).
>     * The observation made above means using the pre-trained word representations (as prior language knowledge) helps the language acquisition from scratch but does not solve the composition generalization challenge.

---

> > ### Author Response · Authors · 2020-11-23
> > **For AnonReviewer4 (C3-C4): Thank you for your review**
> >
> > * **C3 - Misc**:
> >   * **C3.1 - Number of objects (visual tokens) and words (lingual tokens) in an image**
> >     |object_num/img|2|3|4|>=5|total|
> >     |---|---|---|---|---|---|
> >     |samples (%)|60.4%|29.7%|7.7%|2.1%|100.0%|
> >
> >     |words_num/sample|2|3|4|5|6|>=7|total|
> >     |---|---|---|---|---|---|---|---|
> >     |samples (%)|43.6%|28.4%|16.2%|5.6%|3.3%|2.9%|100.0%
> >
> >   * **C3.2 - Figure 3 Attention over visual**
> >     * Yes. Besides image regions, the visual representations of full images provide more information about hand-object interaction or context (similar to the setting in multimodal BERTs [2]). And it could be improved or replaced by using further fine-grained visual annotations (e.g. motion vectors).
> >   * **C.3.3 - Section 3.4**
> >     * Each unique composition was annotated by 3 people and labeled by the majority.
> >   * **C3.4 - Reasoning module**
> >     * The results are shown in the appendix section (e.g., peel carrot = learned operations on [peel potato, stir carrot, cut potato]). And the reasoning module with NAC (Neural Accumulator) is used for learning the arithmetic operations. Our core idea is to disentangle the word semantics and analogy operations during analogy reasoning for compositions.
> >
> > * **C4 - Minor concerns**
> >   * **Section 3.4:** We corrected all the grammatical errors.
> >
> > References:
> > * [1] Vaswani, Ashish, et al. "Attention is all you need." NeurIPS. 2017.
> > * [2] Lu, Jiasen, et al. "Vilbert: Pretraining task-agnostic visiolinguistic representations for vision-and-language tasks." NeurIPS. 2019.

---

> > ### Comment · AnonReviewer1 · 2020-11-24
> > **3 runs or 5?**
> >
> > In your comments to me, you mentioned running each model 5 times, not 3. I agree that standard deviations should be shown. And again, with such slim performance margins, statistical significance testing would go a long way, especially if the word "significant" is being used to describe the differences.

---

> > > ### Author Response · Authors · 2020-11-25
> > > **Thank you for your comments**
> > >
> > > Further responses to AnonReviewer1 are included in the answers for "AnonReviewer1 - Response".
> > > * About randomness effects: It should be "5 times", the typo was corrected.
> > > * About performance improvements: It should be clear that the mentioned "performance improvements" by AnonReviewer4 - C2.1 is not the "performance margins" that AnonReviewer1 mentioned in AnonReviewer1 - Q8. The former means the prediction accuracy results of the Task Evaluation (Table1), while the latter means the validity accuracy results of the Validity Test Evaluation (Table 2). In fact, the prediction accuracy improvements are not slim (as shown in Table1), and the validity accuracies of all baselines are good (as shown in Table2) but ours is better.
> > >   * Prediction Accuracy Improvements for Table 1:
> > >     |Evaluation (new composition)     |prediction |accuracy|
> > >     |---|---|---|
> > >     |our mode v.s. best baseline|top1|top5|
> > >     |Improvements |+2.03%|+4.45%|
> > >     |Relative Improvements |+34%|+12%|
> > >   * Validity Accuracy Comparison with Improvements for Table2:
> > >     |Evaluation (new composition)|validity accuracy (improvement)|
> > >     |---|---|
> > >     |BERT(w/o vision) |81.59% (+4.42%)|
> > >     |Multimodal BERT |84.78% (+1.23%)|
> > >     |ARTNet|86.01%|

---

### Official Review · AnonReviewer2 · 2020-10-28
**Recommendation: accept**

**Rating:** 7
**Confidence:** 4

**Review:**

Summary:

This paper explores the problem of generalizing to novel combinations of verbs and nouns in a task for captioning video stills from videos about cooking. The paper introduces a new dataset based off of EPIC-Kitchens (Damen et al. 2018) which masks out verbs and nouns and splits the evaluation data into seen combinations of verb/noun pairs and unseen combinations of verb/noun pairs, challenging a model to generate captions for pairs which were not seen during training.

The paper also proposed a model, ARTNet, to address this task. The model works by first finding nearest neighbor examples in the training data that are similar in (masked) caption and image, then embedding these similar examples into an "analogical embedding", which is used to condition a network that fills in the masked verb and noun in the target example.

*************

Reason for score:

The proposed dataset is interesting, and the proposed model works well on it. However, I have some concerns about the dataset's construction and suggestions for additional experiments.

*************

Strengths:

1. The proposed model performs well on the proposed task.
1. The baselines seem solid.
1. Includes experiments on different amounts of training data.

*************

Weaknesses:

1. Some missing citations for compositional reasoning on novel images/visual contexts, including datasets like VQA-CP (Agrawal et al. 2017), CLEVR CoGent (Johnson et al. 2017), Room to Room (Anderson et al. 2018), and ALFRED (Shridhar et al. 2020).
1. Would be good to see experiments with a differing number of images in the random pool -- how does this affect performance to have a larger or smaller pool? There seems there would be a tradeoff between accuracy and computing resources necessary to run inference, but what exactly is that tradeoff? Similarly, would be good to see a similar analysis for the number of top candidate examples that are passed to the ARN module from the AMM module.
1. Many terms in the model section should be defined more formally, e.g., "candidate analogy composition", "ordered constituents in a composition", "pars(ing) each sample into candidate analogy compositions", "multimodal resulting set of pairs / analogy pairs", "linguistic clues".

*************

Questions:

1. How does the split for EPIC-Kitchens handle synonyms of verbs? It seems to split based off of verb/noun combinations, but what about things that are (at least partially) synonyms such as "cut", "slice", "chop"? This also seems to be relevant for evaluation -- is it strict for synonyms? (I am looking at the case study, which proposes "slice" squash, "chop" squash", and "cut" squash, all of which seem reasonable to me.)

---

> ### Author Response · Authors · 2020-11-23
> **For AnonReviewer2: Thank you for your review**
>
> We appreciate your comments in recognizing several of the novel ideas and thorough experiments. We address all of your concerns and questions below.
>
> * **W1- Some missing citations:** We added these references for compositional reasoning on novel images/visual contexts.
>
> * **W2 - Experiments:** We added the experiments for the pool size and the numbers of top candidate reference samples and analyzed the performance effects in the revision.
>   * According to the new experiment, the performances slightly increase before and stable after the random pool size reaches the recommended size of 200. A larger pool won’t affect the performance significantly (the performance changes of 200, 300, and 400 are below 0.5%).
>
>     |Evaluation      |new |composition|seen |composition|
>     |---|---|---|---|---|
>     |Accuracy(%)   |top1  |top5    |top1	|top5 |
>     |ref_pool=100  |7.45   |39.60  |21.73	|55.25|
>     |ref_pool=200  |8.00   |41.08  |22.25	|57.74|
>     |ref_pool=300  |7.78   |40.63  |22.13	|57.31|
>     |ref_pool=400  |7.83   |40.98  |21.34	|57.15|
>    *The experiment results for top candidate reference samples refer to the review answer for AnonReviewer4 (“C2.2 - Choice of K in Top-K”).
> * **W3 - Terms in the model section:** We clarified and corrected the terms according to your comments.
>   * “candidate analogy composition”  and "multimodal resulting set of pairs/analogy pairs": They mean the word or image region token pairs for analogy (called analogy pairs) which were collected from top reference samples but still need to be selected again by multimodal analogy reasoning.
>   * “ordered constituents in a composition”: We changed it to “ordered words in a composition” (e.g. The words in “cut apple” are ordered, and different from “apple cut”).
>   * “parsing each sample into candidate analogy compositions”: It means extracting candidate analogy compositions (or analogy pairs) from the multimodal token sequence of each sample
>   * "linguistic clues": It means language-based knowledge
>
> * **Q1 - Synonyms of verbs:** Following the same setting of the EPIC-Kitchens dataset [1], we don’t use/have synonym annotations for verbs and verb-noun composition splits. That means distinct verb-noun combinations would be strictly treated as different compositions even though some of them have synonyms. This is more strict than considering synonyms for evaluation. All the comparisons in our paper used the same setting for fairness.
>
> References:
> * [1] Dima Damen et al., Scaling egocentric vision: The epic-kitchens dataset. ECCV, 2018.

---

### Decision · Program_Chairs · 2021-01-07
**Final Decision**

**Decision:**

Reject

**Comment:**

The authors propose a new dataset and compositional task based on the EPIC Kitchens dataset.  The goal is to test novel compositions and to build a transformer based network specifically for this inference (by analogy). Specifically, the analogy here references the use of nearest neighbors in the dataset.  There are a lot of concerns raised by reviewers which require a large number of changes to the presentation of the manuscript and they are not at present convinced by the current setup or experiments. Explicitly motivating which pretraining methods do or do not violate which aspects of composition and what role other factors like synonymy play in generalization is necessary. Several aspects of the claims made in the paper and in the discussion are big claims that require substantial discussion and analysis (e.g. the surprising weakness of pretrained models) which the reviewers do not feel can be so easily explained away (e.g. by domain shift).